# Visual Concept-Metaconcept Learning

**Chi Han**[*]
MIT CSAIL and IIIS, Tsinghua University

**Jiayuan Mao**[*]
MIT CSAIL

**Chuang Gan**
MIT-IBM Watson AI Lab

**Joshua B. Tenenbaum**
MIT BCS, CBMM, CSAIL

**Jiajun Wu**
MIT CSAIL

## Abstract

Humans reason with concepts and metaconcepts: we recognize *red* and *green* from visual input; we also understand that they *describe the same property of objects* (i.e., the color). In this paper, we propose the visual concept-metaconcept learner (VCML) for joint learning of concepts and metaconcepts from images and associated question-answer pairs. The key is to exploit the bidirectional connection between visual concepts and metaconcepts. Visual representations provide grounding cues for predicting relations between unseen pairs of concepts. Knowing that red and green *describe the same property of objects*, we generalize to the fact that cube and sphere also *describe the same property of objects*, since they both categorize the shape of objects. Meanwhile, knowledge about metaconcepts empowers visual concept learning from limited, noisy, and even biased data. From just a few examples of *purple cubes* we can understand a new color *purple*, which resembles the hue of the cubes instead of the shape of them. Evaluation on both synthetic and real-world datasets validates our claims.

## 1 Introduction

Learning to group objects into concepts is an essential human cognitive process, supporting compositional reasoning over scenes and sentences. To facilitate learning, we have developed metaconcepts to describe the abstract relations between concepts [Speer et al., 2017, McRae et al., 2005]. Learning both concepts and metaconcepts involves categorization at various levels, from concrete visual attributes such as *red* and *cube*, to abstract relations between concepts, such as *synonym* and *hypernym*. In this paper, we focus on the problem of learning visual concepts and metaconcepts with a linguistic interface, from looking at images and reading paired questions and answers.

Figure 1a gives examples of concept learning and metaconcept learning in the context of answering visual reasoning questions and purely textual questions about metaconcepts. We learn to distinguish *red* objects from *green* ones by their hues, by looking at visual reasoning (type I) examples. We also learn metaconcepts, e.g., red and green *describe the same property of objects*, by reading metaconcept (type II) questions and answers.

Concept learning and metaconcept learning help each other. Figure 1b illustrates the idea. First, metaconcepts enable concept learning from limited, noisy, and even biased examples, with generalization to novel compositions of attributes at test time. Assuming only a few examples of red cubes with the label *red*, the visual grounding of the word *red* is ambiguous: it may refer to the hue red or cube-shaped objects. We can resolve such ambiguities knowing that red and green *describe the same property of objects*. During test (Figure 1b-I), we can then generalize to red cylinders. Second, concept learning provides visual cues for predicting relations between unseen pairs of concepts. After learning that red and green *describe the same property of objects*, one may hypothesize a

---

First two authors contributed equally. Work was done when Chi Han was a visiting student at MIT CSAIL.
Project Page: `http://vcml.csail.mit.edu`.

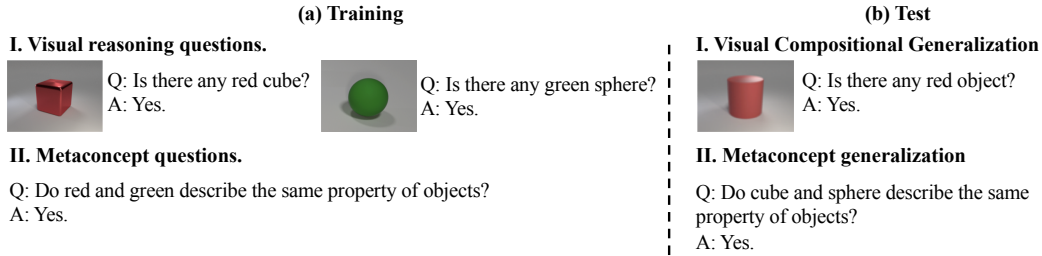

Figure 1: Our model learns concepts and metaconcepts from images and two types of questions. The learned knowledge helps visual concept learning (generalizing to unseen visual concept compositions, or to concepts with limited visual data) and metaconcept generalization (generalizing to relations between unseen pairs of concepts.)

generalization of the notion "same property" to *cube* and *sphere*, since they both categorize objects by their shapes (in comparison to red and green both categorizing the hue, sees Figure 1b-II).

Based on this observation, we propose the visual concept-metaconcept learner (VCML, see Figure 2) for joint learning of visual concepts (*red* and *cube*) and metaconcepts (e.g., red and green *describe the same property of objects*). VCML consists of three modules: a visual perception module extracting an object-based representation of the input image, a semantic parser translating the natural language question into a symbolic program, and a neuro-symbolic program executor executing the program based on the visual representation to answer the question. VCML bridges the learning of visual concepts and metaconcepts with a latent vector space. Concepts are associated with vector embeddings, whereas metaconcepts are neural operators that predict relations between concepts.

Both concept embeddings and metaconcept operators are learned by looking at images and reading question-answer pairs. Our training data are composed of two parts: 1) questions about the visual grounding of concepts (e.g., *is there any red cube?*), and 2) questions about the abstract relations between concepts (e.g., *do red and green describe the same property of objects?*).

VCML generalizes well in two ways, by learning from the two types of questions. It can successfully categorize objects with new combinations of visual attributes, or objects with attributes with limited training data (Figure 1b, type I); it can also predict relations between unseen pairs of concepts (Figure 1b, type II). We present a systematic evaluation on both synthetic and real-world images, with a focus on learning efficiency and strong generalization.

## 2   Related Work

Learning visual concepts from language or other forms of symbols, such as class labels or tags, serves as a prerequisite for a broad set of downstream visual-linguistic applications, including cross-modal retrieval [Kiros et al., 2014], caption generation [Karpathy and Fei-Fei, 2015], and visual-question answering [Malinowski et al., 2015]. Existing literature has been focused on improving visual concept learning by introducing new representations [Wu et al., 2017], new forms of supervisions [Johnson et al., 2016, Ganju et al., 2017], new training algorithms [Faghri et al., 2018, Shi et al., 2018], structured and geometric embedding spaces [Ren et al., 2016, Vendrov et al., 2016], and extra knowledge base [Thoma et al., 2017].

Our model learns visual concepts by reasoning over question-answer pairs. Prior works on visual reasoning have proposed to use end-to-end neural networks for jointly learning visual concepts and reasoning [Malinowski et al., 2015, Yang et al., 2016, Xu and Saenko, 2016, Andreas et al., 2016, Gan et al., 2017, Mascharka et al., 2018, Hudson and Manning, 2018, Hu et al., 2018], whereas some recent papers [Yi et al., 2018, Mao et al., 2019, Yi et al., 2019] attempt to disentangle visual concept learning and reasoning. The disentanglement brings better data efficiency and generalization.

In this paper, we study the new challenge of incorporating metaconcepts, i.e., relational concepts about concepts, into visual concept learning. Beyond just learning from questions regarding visual scenes (e.g., is there any red cubes?), our model learns from questions about metaconcepts (e.g., do red and yellow describe the same property of objects?). Both concepts and metaconcepts are learned with a unified neuro-symbolic reasoning process. Our evaluation focuses on revealing bidirectional connections between visual concept learning and metaconcept learning. Specifically, we study

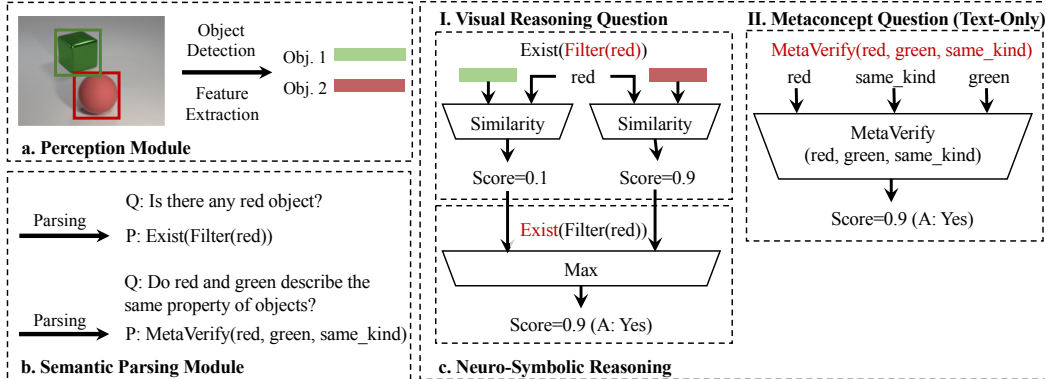

Figure 2: The Visual Concept-Metaconcept Learner. The model comprises three modules: (a) a perception module for extracting object-based visual representations, (b) a semantic parsing module for recovering latent programs from natural language, and (c) a neuro-symbolic reasoning module that executes the program to answer the question.

both visual compositional generalization and metaconcept generalization (i.e., relational knowledge generalization).

Visual compositional generalization focuses on exploiting the compositional nature of visual categories. For example, the compositional concept *red cube* can be factorized into two concepts: *red* and *cube*. Such compositionality suggests the ability to generalize to unseen combination of concepts: e.g., from *red cube* and *yellow sphere* to *red sphere*. Many approaches towards compositional visual concept learning have been proposed, including compositional embeddings [Misra et al., 2017], neural operators [Nagarajan and Grauman, 2018], and neural module networks [Purushwalkam et al., 2019]. In this paper, we go one step further towards compositional visual concept learning by introducing metaconcepts, which empower learning from biased data. As an example, by looking at only examples of *red cubes*, our model can recover the visual concept *red* accurately, as a category of chromatic color, and generalizes to unseen compositions such as *red sphere*.

Generalizing from known relations between concepts to unseen pairs of concepts can be cast as an instance of relational knowledge base completion. The existing literature has focused on learning vector embeddings from known knowledge [Socher et al., 2013, Bordes et al., 2013, Wang et al., 2014], recovering logic rules between metaconcepts [Yang et al., 2017], and learning from corpora [Lin et al., 2017]. In this paper, we propose a visually-grounded metaconcept generalization framework. This allows our model, for example, to generalize from *red and yellow describe the same property of objects* to *green and yellow also describe the same property of objects* by observing that both *red* and *green* classify objects by their hues based on vision.

## 3 Visual Concept-Metaconcept Learning

We present the visual concept-metaconcept learner (VCML), a unified framework for learning visual concepts and metaconcepts by reasoning over questions and answers about scenes. It answers both visual reasoning questions (e.g., is there any red object?) and text-only metaconcept questions (e.g., do red and green describe the same property of objects?) with a unified neuro-symbolic framework. VCML comprises three modules (Figure 2):

- A *perception* module (Figure 2a) for extracting an object-based representation of the scene, where each object is represented as a vector embedding of a fixed dimension.

- A *semantic parsing* module (Figure 2b) for translating the input question into a symbolic executable program. Each program consists of hierarchical primitive functional modules for reasoning. Concepts are represented as vector embeddings in a latent vector space, whereas metaconcepts are small neural networks predicting relations between concepts. The concept embeddings are associated with the visual representations of objects.

- A *neuro-symbolic reasoning* module (Figure 2c) for executing the program to answer the question, based on the scene representation, concept embeddings, and metaconcept operators. During

Table 1: Our extension of the visual-reasoning DSL [Johnson et al., 2017, Hudson and Manning, 2019], including one new primitive function for metaconcepts.

| MetaVerify | |
|---|---|
| **Signature** | Concept, Concept, MetaConcept $\longrightarrow$ Bool |
| **Semantics** | Returns whether two input concepts have the specified metaconcept relation. |
| **Example** | MetaVerify(Sphere, Ball, Synonym) $\longrightarrow$ True |

training, it also receives the groundtruth answer as the supervision and back-propagates the training signals.

## 3.1 Motivating Examples

In Figure 2, we illustrate VCML by walking through two motivating examples of visual reasoning and metaconcept reasoning.

**Visual reasoning.** Given the input image, the perception module generates object proposals for two objects and extracts vector representations for them individually. Meanwhile, the question *is there any red object* will be translated by a semantic parsing module into a two-step program: `Exist(Filter(red))`. The neuro-symbolic reasoning module executes the program. It first computes the similarities between the concept embedding *red* and object embeddings to classify both objects. Then, it answers the question by checking whether a red object has been filtered out.

**Metaconcept questions.** Metaconcept questions are text-only. Each of the metaconcept questions queries a metaconcept relation between a pair of concepts. Specifically, consider the question *do red and green describe the same property of objects*. We denote two concepts are related by a *same_kind* metaconcept if they describe the same property of objects. To answer this question, we first run the semantic parsing module to translate it into a symbolic program: `MetaVerify(red, green, same_kind)`. The neuro-symbolic reasoning module answers the question by inspecting the latent embeddings of two concepts (*red* and *green*) with the metaconcept operator of *same_kind*.

## 3.2 Model Details

**Perception module.** Given the input image, VCML builds an object-based representation of the scene. This is done by using a Mask R-CNN [He et al., 2017] to generate object proposals, followed by a ResNet-34 [He et al., 2015] to extract region-based feature representations of individual objects.

**Semantic parsing module.** The semantic parsing module takes the question as input and recovers a latent program. The program has a hierarchical structure of primitive operations such as filtering out a set of objects with a specific concept, or evaluating whether two concepts are *same_kind*.

The domain-specific language (DSL) of the semantic parser extends the DSL used by prior works on visual reasoning [Johnson et al., 2017, Hudson and Manning, 2019] by introducing new functional modules for metaconcepts (see Table 1).

**Concept embeddings.** The concept embedding space lies at the core of this model. It is a joint vector embedding space of object representations and visual concepts. Metaconcepts operators are small neural networks built on top of it. Figure 3 gives a graphical illustration of the embedding space.

We build the concept embedding space drawing inspirations from the order-embedding framework [Ivan Vendrov, 2016] and its extensions [Lai and Hockenmaier, 2017, Vilnis et al., 2018]. By explicitly defining a partial order or entailment probabilities between vector embeddings, these models are capable of learning a well structured embedding space which captures certain kinds of relations among the concept embeddings.

In VCML, we have designed another probabilistic order embedding in a high-dimensional space $\mathbb{R}^N$. Each object and each visual concept (e.g., *red*) is associated with an embedding vector $\mathbf{x} \in \mathbb{R}^N$. This vector defines a half-space $V(\mathbf{x}) = \{\mathbf{y} \in \mathbb{R}^N \mid (\mathbf{y} - \mathbf{x})^T \mathbf{x} > 0\}$. We assume a standard normal distribution $\mathcal{N}(\mathbf{0}, \mathbf{I})$ distrbuted over the whole space. We further define a denotational probability for each entity $a$, which can be either an object or a concept, with the associated embedding vector $\mathbf{x}_a$ as

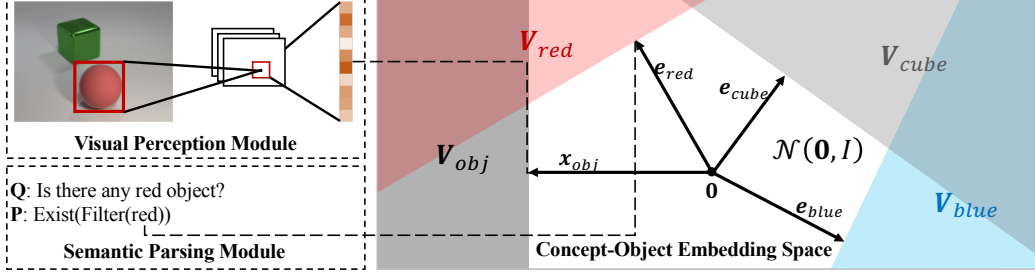

Figure 3: A graphical illustration of our concept-metaconcept embedding space. Each concept or object is embedded as a high-dimensional vector, which is associated with a half-space supported by this vector. Each metaconcept is associated with a multi-layer perceptron as a classifier. The programs are executed based on the scene representation, concept embeddings and metaconcept operators.

the measure of $V_a = V(\mathbf{x}_a)$ over this distribution:

$$\Pr(a) = \text{Vol}_{\mathcal{N}(\mathbf{0},\mathbf{I})}(V_a) = \int_{\mathbf{z} \in V_a} \frac{1}{\sqrt{2\pi}} e^{-\frac{1}{2}\|\mathbf{z}\|_2^2} d\mathbf{z} = \frac{1}{2}[1 - \text{erf}(\frac{\|\mathbf{x}\|_2}{\sqrt{2}})]$$

Similarly, the joint probability of two entites $a$, $b$ can be computed as the measure of the intersection of their half-spaces:

$$\Pr(a,b) = \text{Vol}_{\mathcal{N}(\mathbf{0},\mathbf{I})}(V_a \cap V_b) = \int_{\mathbf{z} \in V_a \cap V_b} \frac{1}{\sqrt{2\pi}} e^{-\frac{1}{2}\|\mathbf{z}\|_2^2} d\mathbf{z}$$

We can therefore define the entailment probability between two entities as

$$\Pr(b \mid a) = \frac{\Pr(a,b)}{\Pr(a)} = \frac{\text{Vol}_{\mathcal{N}(\mathbf{0},\mathbf{I})}(V_b \cap V_a)}{\text{Vol}_{\mathcal{N}(\mathbf{0},\mathbf{I})}(V_a)}$$

This entailment probability can then be used to give (asymmetric) similarity scores between objects and concepts. For example, to classify whether an object $o$ is *red*, we compute $\Pr(\text{object } o \text{ is } red) = \Pr(red \mid o)$.

**Metaconcept operators** For simplicity, we only consider metaconcepts defined over a pair of concepts, such as *synonym* and *same_kind*. Each metaconcept is associated with a multi-layer perceptron (e.g., $f_{\text{synonym}}$ for the metaconcept *synonym*). To classify whether two concepts (e.g., *red* and *cube*) are related by a metaconcept (e.g., *synonym*), we first compute several denotational entailment probabilities between them. Mathematical, we define two helper functions $g_1$ and $g_2$:

$$g_1(a,b) = \text{logit}(\Pr(a \mid b)), \quad g_2(a,b) = \ln \frac{\Pr(a,b)}{\Pr(a)\Pr(b)} \ ,$$

where $\text{logit}(\cdot)$ is the logit function. These values are then fed into the perceptron to predict the relation

$$\text{MetaVerify}(red, cube, synonym) = f_{synonym}(g_1(red, cube), g_1(cube, red), g_2(red, cube)) \ .$$

**Neuro-symbolic reasoning module.** VCML executes the program recovered by the semantic parsing module with a neuro-symbolic reasoning module [Mao et al., 2019]. It contains a set of deterministic functional modules and does not require any training. The high-level idea is to relax the boolean values during execution into soft scores ranging from 0 to 1. Illustrated in Figure 2, given a scene of two objects, the result of a `Filter(red)` operation is a vector of length two, where the $i$-th element denotes the probability whether the $i$-th object is red. The `Exist(·)` operation takes the max value of the input vector as the answer to the question. The neuro-symbolic execution allows us to disentangle visual concept learning and metaconcept learning from reasoning. The derived answer to the question is fully differentiable with respect to the concept-metaconcept representations.

### 3.3 Training

There are five modules to be learned in VCML: the object proposal generator, object representations, the semantic parser, concept embeddings, and metaconcept operators. Since we focus on the concept-metaconcept learning problem, we assume the access to a pre-trained object proposal generator and

a semantic parser. The ResNet-34 model for extracting object features is pretrained on ImageNet [Deng et al., 2009], and gets finetuned during training. Unless otherwise stated, the concepts and metaconcepts are learned with an Adam optimizer [Kingma and Ba, 2015] based on learning signals back-propagated from the neuro-symbolic reasoning module. We use a learning rate of 0.001 and a batch size of 10.

# 4 Experiments

The experiment section is organized as follows. In Section 4.1 and Section 4.2, we introduce datasets we used and the baselines we compare our model with, respectively. And then we evaluate the generalization performance of various models from two perspectives. First, in Section 4.3, we show that incorporating metaconcept learning improves the data efficiency of concept learning. It also suggests solutions to compositional generalization of visual concepts. Second, in Section 4.4, we show how concept grounding can help the prediction of metaconcept relations between unseen pairs of concepts.

All results in this paper are the average of four different runs, and $\pm$ in results denotes standard deviation.

## 4.1 Dataset

We evaluate different models on both synthetic (CLEVR [Johnson et al., 2017]) and natural image (GQA [Hudson and Manning, 2019], CUB [Wah et al., 2011]) datasets. To have a fine-grained control over question splits, we use programs to generate synthetic questions and answers, based on the ground-truth annotations of visual concepts.

The CLEVR dataset [Johnson et al., 2017] is a diagnostic dataset for visual reasoning. It contains synthetic images of objects with different sizes, colors, materials, and shapes. There are a total of 22 visual concepts in CLEVR (including concepts and their synonyms). We use a subset of 70K images from the CLEVR training split for learning visual concepts and metaconcepts. 15K images from the validation split are used during test.

The GQA dataset [Hudson and Manning, 2019] contains visual reasoning questions for natural images. Each image is associated with a scene graph annotating the visual concepts in the scene. It contains a diverse set of 2K visual concepts. We truncate the long-tail distribution of concepts by selecting a subset of 584 most frequent concepts. We use 75K images for training and another 10K images for test. In GQA, we use the ground-truth bounding boxes provided in the dataset. Attribute labels are not used for training our model or any baselines.

The CUB dataset [Wah et al., 2011] contains around 12K images of 200 bird species. Each image contains one bird, with classification and body part attribute annotations (for the *meronym* metaconcept). We use a subset of 8K images for training and another 1K images for test.

In addition to the visual reasoning questions, we extend the datasets to include metaconcept questions. These questions are generated according to external knowledge bases. For CLEVR, we use the original ontology. For GQA, we use the *synsets* and *same_kind* relations from the WordNet [Miller, 1995]. For CUB, we extend the ontology with 166 bird taxa of higher categories (families, genera, etc.), and use the hypernym relations from the eBird Taxonomy [Sullivan BL, 2009] to build a hierarchy of all the taxonomic concepts. We also use the attribute annotations in CUB.

## 4.2 Baseline

We compare VCML with the following baselines: general visual reasoning frameworks (GRU-CNN and MAC), concept learning frameworks (NS-CL), and language-only frameworks (GRU and BERT).

**GRU-CNN.** The GRU-CNN baseline is a simple baseline for visual question answering [Zhou et al., 2015]. It consists of a ResNet-34 [He et al., 2015] encoder for images and a GRU [Cho et al., 2014] encoder for questions. To answer a question, image features and question features are concatenated, followed by a softmax classifier to predict the answer. For text-only questions, we use only question features.

**MAC.** We replicate the result of the MAC network [Hudson and Manning, 2018] for visual reasoning, which is a dual attention-based model. For text-only questions, the MAC network takes a blank image as the input.

Table 2: The metaconcept *synonym* provides abstract-level supervision for concepts. This enables zero-shot learning of novel concepts.

|  | GRU-CNN | MAC | NS-CL | VCML |
|---|---|---|---|---|
| **CLEVR** | $50.0_{\pm0.0}$ | $68.7_{\pm3.8}$ | $80.2_{\pm3.1}$ | $\mathbf{94.1_{\pm4.6}}$ |
| **GQA** | $50.0_{\pm0.0}$ | $49.5_{\pm0.2}$ | $49.3_{\pm0.6}$ | $\mathbf{50.5_{\pm0.1}}$ |

Table 3: The metaconcept *same_kind* helps the model learn from biased data and generalize to novel combinations of visual attributes.

|  | GRU-CNN | MAC | NS-CL | VCML |
|---|---|---|---|---|
| **CLEVR-200** | $50.0_{\pm0.0}$ | $94.2_{\pm3.3}$ | $98.5_{\pm0.3}$ | $\mathbf{98.9_{\pm0.2}}$ |
| **CLEVR-20** | $50.0_{\pm0.0}$ | $79.7_{\pm2.6}$ | $\mathbf{95.7_{\pm0.0}}$ | $95.1_{\pm1.6}$ |

Table 4: The metaconcept *hypernym* enables few-shot learning of new concepts.

|  | GRU-CNN | MAC | NS-CL | VCML |
|---|---|---|---|---|
| **CUB** | $50.0_{\pm0.0}$ | $70.8_{\pm3.4}$ | $80.0_{\pm2.3}$ | $\mathbf{80.2_{\pm1.7}}$ |

Table 5: Application of VCML on Referential Expression task on CLEVR dataset

|  | #Train | w/. | w/o. |
|---|---|---|---|
| Ref. Expr. | 10K | $\mathbf{74.9_{\pm0.1}}$ | $73.8_{\pm1.7}$ |
|  | 1K | $\mathbf{59.7_{\pm0.2}}$ | $51.6_{\pm2.6}$ |

**NS-CL.** The NS-CL framework is proposed by Mao et al. [2019] for learning visual concepts from visual reasoning. Similar to VCML, it works on object-based representations for scenes and program-like representations for questions. We extend it to support functional modules of metaconcepts. They are implemented as a two-layer feed forward neural network that takes concept embeddings as input.

**GRU (language only).** We include a language-only baseline that uses a GRU [Cho et al., 2014] to encode the question and a softmax layer to predict the answer. We use pre-trained GloVe [Pennington et al., 2014] word embeddings as concept embeddings. We fix word embeddings during training, and only train GRU weights on the language modeling task on training questions.

**BERT.** We also include BERT [Jacob Devlin, 2019] as a language-only baseline. Two variants of BERT are considered here. Variant I encodes the natural language question with BERT and uses an extra single-layer perceptron to predict the answer. Vartiant II works with a pre-trained semantic parser as the one in VCML. To predict metaconcept relations, it encodes concept words or phrases into embedding vectors, concatenates them, and applies a single-layer perceptron to answer the question. During training, the parameters of the BERT encoders are always fixed.

### 4.3 Metaconcepts Help Concept Learning

Metaconcepts help concept learning by providing extra supervision at an abstract level. Three types of generalization tests are studied in this paper. First, we show that the metaconcept *synonym* enables the model to learn a novel concept without any visual examples (i.e., zero-shot learning). Second, we demonstrate how the metaconcept *same_kind* supports learning from biased visual data. Third, we evaluate the performance of few-shot learning with the support of the metaconcept *hypernym*. Finally, we provide extra results to demonstrate that metaconcepts can improve the overall data-efficiency of visual concept learning. For more examples on the data split, please refer to the supplementary material.

#### 4.3.1 *synonym* Supports Zero-Shot Learning of Novel Concepts

The metaconcept *synonym* provides abstract-level supervision for concept learning. With the visual grounding of the concept *cube* and the fact that *cube* is a synonym of *block*, we can easily generalize to recognize *blocks*. To evaluate this, we hold out a set of concepts $\mathcal{C}_{test}^{syn}$ that are synonyms of other concepts. The training dataset contains synonym metaconcept questions about $\mathcal{C}_{test}^{syn}$ concepts but no visual reasoning questions about them. In contrast, all test questions are visual reasoning questions involving $\mathcal{C}_{test}^{syn}$.

**Dataset.** For the CLEVR dataset, we hold out three concepts out of 22 concepts. For the GQA dataset, we hold out 30 concepts.

**Results.** Quantitative results are summarized in Table 2. Our model significantly outperforms all baselines that are metaconcept-agnostic on the synthetic dataset CLEVR. It also outperforms all other methods on the real-world dataset GQA, but the advantage is smaller. We attribute this result to the complex visual features and the vast number of objects in the real world scenes, which degrade the performance of concept learning. As an ablation, we test the trained model on a validation set which has the same data distribution as the training set. The train-validation gap on GQA (training: 83.1%; validation: 52.3%) is one-magnitude-order larger than the gap on CLEVR (training: 99.4%; validation: 99.4%).

### 4.3.2 *same_kind* Supports Learning from Biased Data

The metaconcept *same_kind* supports visual concept learning from biased data. Here, we focus on biased visual attribute composition in the training set. For example, from just a few examples of *purple cubes*, the model should learn a new color *purple*, which resembles the hue of the cubes instead of the shape of them.

**Dataset.**    We replicate the setting of CLEVR-CoGenT [Johnson et al., 2017], and create two splits of the CLEVR dataset: in split A, all cubes are gray, yellow, brown, or yellow, whereas in split B, cubes are red, green, purple or cyan. In training, we use all the images in split A, together with a few images from split B (which are 200 images from split B in the CLEVR-200 group, and 20 in the CLEVR-20 group, shown in Table 3). During training, we use metaconcept questions to indicate that cube categorizes shapes of objects rather than colors. The held out images that in split B are used for test.

**Results.**    Table 3 shows the results. VCML and NS-CL successfully learn visual concepts from biased data through the concept-metaconcept integration. We also evaluate all trained models on a validation set which has the same data distribution as the training set. We found that most models perform equally well on this validation set (for example, MAC gets 99.1% accuracy in the validation set, while both NSCL and VCML get 99.7%). The contrast between the validation accuracies and test accuracies supports that only a better concept-metaconcept integration contributes to the visual concept learning in such biased settings. Please refer to the supplementary material for more details.

### 4.3.3 *hypernym* Supports Few-Shot Learning Concepts

The abstract-level supervision provided by the metaconcept *hypernym* supports learning visual concepts from limited data. After having learned the visual concept *Sterna*, and the fact that *Sterna* is a *hypernym* of *Arctic Tern*, we can narrow down the possible visual grounding of *Arctic Tern*. This helps the model to learn the concept *Arctic Tern* with fewer data.

**Dataset.**    We select 91 out of all 366 taxonomic concepts in CUB dataset [Wah et al., 2011] as $\mathcal{C}_{test}^{hyp}$. In the training set, there are only 5 images per concept for the concepts in $\mathcal{C}_{test}^{hyp}$. In contrast, each of the other concepts are associated with around 40 images during training. We evaluate different models with visual reasoning questions about concepts in $\mathcal{C}_{test}^{hyp}$, based on the held-out images. All visual reasoning questions are generated based on the class labels of images.

**Results.**    The results in Table 4 show that our model outperforms both GRU-CNN and MAC. NS-CL [Mao et al., 2019], augmented with metaconcept operators, also achieves a comparable performance as VCML. To further validate the effectiveness of extra metaconcept knowledge, we also test the performance of different models when the metaconcept questions are absent. Almost all models show degraded performance to various degrees (for example, MAC gets a test accuracy of 60.4%, NS-CL gets 79.6%, and VCML gets 78.5%).

### 4.3.4 Application of concept embeddings to downstream task

We supplement extra results on the CLEVR referential expression task. This task is to select out a specific object from a scene given a description (e.g., the red cube). We compare VCML with and without metaconcept information using Recall@1 for referential expressions.

**Results.**    Table 5 suggests that the metaconcept information significantly improves visual concept learning in low-resource settings, using only 10K or even 1K visually grounded questions. We found that as the number of visually-grounded questions increases, the gap between training with and without metaconcept questions gets smaller. We conjecture that this is an indication of the model relying more on visual information instead of metaconcept knowledge when there is larger visual reasoning dataset.

### 4.4 Concepts Help Metaconcept Generalization

Concept learning provides visual cues for predicting the relations between unseen pairs of concepts. We quantitatively evaluate different models by their accuracy of predicting metaconcept relations between unseen pairs. Four representative metaconcepts are studied here: *synonym*, *same_kind*, *hypernym* and *meronym*.

Table 6: Metaconcept generalization evaluation on the CLEVR, GQA and CUB dataset. (Two variants of BERT are shown here; see Section 4.2 for details.)

| | | Q.Type | GRU (Lang. Only) | GRU-CNN | BERT (Variant I ; Variant II) | NS-CL | VCML |
|---|---|---|---|---|---|---|---|
| **CLEVR** | Synonym | 50.0 | $66.3_{\pm 1.4}$ | $60.9_{\pm 10.6}$ | $76.2_{\pm 10.2}$ ; $80.2_{\pm 16.1}$ | $\mathbf{100.0_{\pm 0.0}}$ | $\mathbf{100.0_{\pm 0.0}}$ |
| | Same-kind | 50.0 | $64.7_{\pm 5.1}$ | $61.5_{\pm 6.6}$ | $75.4_{\pm 5.4}$ ; $80.1_{\pm 10.0}$ | $92.3_{\pm 4.9}$ | $\mathbf{99.3_{\pm 1.0}}$ |
| **GQA** | Synonym | 50.0 | $80.8_{\pm 1.0}$ | $76.2_{\pm 0.8}$ | $76.2_{\pm 2.4}$ ; $83.1_{\pm 1.5}$ | $81.2_{\pm 2.8}$ | $\mathbf{91.1_{\pm 1.7}}$ |
| | Same-kind | 50.0 | $56.3_{\pm 2.3}$ | $57.3_{\pm 5.3}$ | $59.5_{\pm 2.7}$ ; $68.2_{\pm 4.0}$ | $66.8_{\pm 4.1}$ | $\mathbf{69.1_{\pm 1.7}}$ |
| **CUB** | Hypernym | 50.0 | $74.3_{\pm 5.2}$ | $76.7_{\pm 8.8}$ | $75.6_{\pm 1.2}$ ; $61.7_{\pm 10.3}$ | $80.1_{\pm 7.3}$ | $\mathbf{94.8_{\pm 1.3}}$ |
| | Meronym | 50.0 | $80.1_{\pm 5.9}$ | $78.1_{\pm 4.8}$ | $63.1_{\pm 3.2}$ ; $72.9_{\pm 9.9}$ | $\mathbf{97.7_{\pm 1.1}}$ | $92.5_{\pm 1.0}$ |

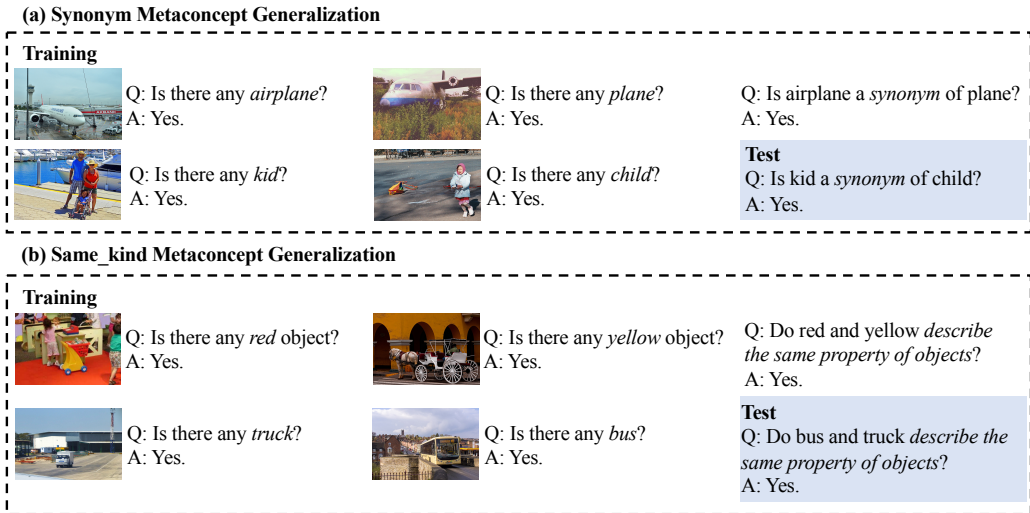

Figure 4: Data split used for metaconcept generalization tests. The models are required to leverage the visual analogy between concepts to predict metaconcepts about unseen pairs of concepts (shown in blue). More details for other metaconcepts can be found in the supplementary material.

**Dataset.** Figure 4 shows the training-test split for the metaconcept generalization test. For each metaconcept, a subset of concepts $\mathcal{C}_{test}^{[\text{metaconcept}]\_gen}$ are selected as test concepts . The rest concepts form the training concept set $\mathcal{C}_{train}^{[\text{metaconcept}]\_gen}$. Duing training, only metaconcept questions with both queried concepts in $\mathcal{C}_{train}^{[\text{metaconcept}]\_gen}$ are used. Metaconcept questions with both concepts in $\mathcal{C}_{test}^{[\text{metaconcept}]\_gen}$ are used for test. Models should leverage the visual grounding of concepts to predict the metaconcept relation between unseen pairs.

**Results.** The results for metaconcept generalization on the three datasets are summarized in Table 6. The question type baseline (shown as Q. Type) is the best-guess baseline for all metaconcepts. Overall, VCML achieves the best metaconcept generalization, and only shows inferior performance to NS-CL on the *meronym* metaconcept. Note that the NS-CL baseline used here is our re-implementation that augments the original version with similar metaconcept operators as VCML.

## 5 Conclusion

In this paper, we propose the visual concept-metaconcept learner (VCML) for bridging visual concept learning and metaconcept learning (i.e., relational concepts about concepts). The model learns concepts and metaconcepts with a unified neuro-symbolic reasoning procedure and a linguistic interface. We demonstrate that connecting visual concepts and abstract relational metaconcepts bootstraps the learning of both. Concept grounding provides visual cues for predicting relations between unseen pairs of concepts, while the metaconcepts, in return, facilitate the learning of concepts from limited, noisy, and even biased data. Systematic evaluation on the CLEVR, CUB, and GQA datasets shows that VCML outperforms metaconcept-agnostic visual concept learning baselines as well as visual reasoning baselines.

**Acknowledgement.** We thank Jon Gauthier for helpful discussions and suggestions. This work was supported in part by the Center for Brains, Minds and Machines (CBMM, NSF STC award CCF-1231216), ONR MURI N00014-16-1-2007, MIT-IBM Watson AI Lab, and Facebook.

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
