[Supplementary Material]

# Supplementary Material for: Visual Concept-Metaconcept Learning

**Chi Han**[*]
MIT CSAIL and IIIS, Tsinghua University

**Jiayuan Mao**[*]
MIT CSAIL

**Chuang Gan**
MIT-IBM Watson AI Lab

**Joshua B. Tenenbaum**
MIT BCS, CBMM, CSAIL

**Jiajun Wu**
MIT CSAIL

## 1 Dataset Visualization

In this section, we explain in detail the training-test split for the datasets with examples, with images from CLEVR[Johnson et al., 2017], GQA[Hudson and Manning, 2019] and CUB[Wah et al., 2011].

In this paper we consider two perspectives of generalization. First, we demonstrate that, with the help of metaconcept questions, VCML is capable of improving its data efficiency in concept learning as well as generalizing to unseen visual attributes compositions.

**(a) *Synonym* Supports Zero-Shot Learning of Novel Concepts**

**I. Visual reasoning questions**
Q: Is there any *cube*?
A: Yes.

**II. Metaconcept questions**
Q: Is block a *synonym* of cube?
A: Yes.

Q: Is there any *block*?
A: Yes.

**(b) *Same-kind* Supports Learning from Biased Data**

**I. Visual reasoning questions**
Q: Is there any *red cube*?
A: Yes.

Q: Is there any *sphere*?
A: Yes.

**II. Metaconcept questions**
Q: Do cube and sphere *describe the same property of objects*?
A: Yes.

Q: Is there any *cube*?
A: Yes.

**(c) *Hypernym* Supports Few-Shot Learning of Concepts**

**I. Visual reasoning questions**
Q: Is there any *Sterna*?
A: Yes.

Q: Is there any *Arctic Tern*?
A: Yes.

**II. Metaconcept questions**
Q: Is Sterna a *hypernym* of Arctic Tern?
A: Yes.

Q: Is there any *Arctic Tern*?
A: Yes.

**Training**                    **Test (visual reasoning)**

Figure 1: Data split used for tests on "Metaconcepts help concept learning". In training, the visual data are insufficient or even biased for some concepts. The models are then required to leverage the information provided in metaconcept questions in order to correctly reason on visual questions about these concepts (shown in blue).

---

First two authors contributed equally. Work was done when Chi Han was a visiting student at MIT CSAIL.
Project Page: http://vcml.csail.mit.edu.

**(a)** *Synonym* **Metaconcept Generalization**

**Training**

Q: Is there any *airplane*?
A: Yes.

Q: Is there any *plane*?
A: Yes.

Q: Is airplane a *synonym* of plane?
A: Yes.

Q: Is there any *kid*?
A: Yes.

Q: Is there any *child*?
A: Yes.

**Test**
Q: Is kid a *synonym* of child?
A: Yes.

**(b)** *Same-kind* **Metaconcept Generalization**

**Training**

Q: Is there any *red* object?
A: Yes.

Q: Is there any *yellow* object?
A: Yes.

Q: Do red and yellow *describe the same property of objects*?
A: Yes.

Q: Is there any *truck*?
A: Yes.

Q: Is there any *bus*?
A: Yes.

**Test**
Q: Do bus and truck *describe the same property of objects*?
A: Yes.

**(c)** *Hypernym* **Metaconcept Generalization**

**Training**

Q: Is there any *Sterna* ?
A: Yes.

Q: Is there Arctic Tern?
A: Yes.

Q: Is Sterna a *hypernym* of the Arctic Stern?
A: Yes.

Q: Is there any *Lanius*?
A: Yes.

Q: Is there any *Loggerhead shrike*?
A: Yes.

**Test**
Q: Is Lanius a *hypernym* of the Loggerhead Shrike?
A: Yes.

**(d)** *Meronym* **Metaconcept Generalization**

**Training**

Q: Is there any *Ivory Gull*?
A: Yes.

Q: Is there any object with a *white breast*?
A: Yes.

Q: Is the white breast a *meronym* of the Ivory Gull?
A: Yes.

Q: Is there any *Black Tern*?
A: Yes.

Q: Is there any object with a *black bill*?
A: Yes.

**Test**
Q: Is the black bill a *meronym* of the Black Tern?
A: Yes.

Figure 2: Data split used for metaconcept generalization tests. The models are required to leverage the visual analogy between concepts to predict metaconcepts about unseen pairs of concepts (shown in blue). See the main text for details.

Figure 1 shows examples for the three tasks evaluated in this type of generalization. In training, we use visual reasoning datasets for learning the visual grounding of concepts. Furthermore, we provide metaconcept questions to provide abstract-level supervision. In test (shown in blue rectangles), the models are required to correctly reason on visual concepts.

Second, we study how learned visual concepts provide grounding cues to predict metaconcept relations between unseen pairs of concepts. Figure 2 illustrates the training-test splits for four metaconcept generalization tests. In training, we provide visual reasoning questions and a subset of metaconcept questions. In test, the models are required to generalize the learned metaconcepts to unseen pairs of concepts.

## 2 Ablation: *same_kind* Supports Learning from Biased Data

To further quantify the effectiveness of metaconcepts in supporting learning from biased data, we conducted an ablation study. Recall that the training questions are from two sources: all images in the split A, and a small number of images from the split B. We plot the performance of different models by varying the number of training images from the split B.

Three models are tested in Figure 3: VCML, NS-CL [Mao et al., 2019], MAC [Hudson and Manning, 2018]. We also evaluate the performance of VCML if all metaconcept questions are absent, shown as VCML (ablation).

Overall, VCML outperforms two other baselines by a margin when the number of training images from the split B is greater than 3. NS-CL significantly closes the gap when there are more than 10 split B images. However, VCML trained without metaconcept questions also achieves a comparable performance when the number of split B images is large. This suggests that our model outperforms both baselines in utilizing metaconcepts to support learning from biased data.

**Test Accuracy**

Figure 3: Models results under different levels of visual compositional bias in the experiment "*same_kind* Supports Learning from Biased Data". The *y*-axis is the test accuracy in percent, and the *x*-axis is the number of split B images in training. Plots are results for different models/settings. The transparent band denotes $\pm\frac{1}{2}$ standard deviation.