[Reviews · NeurIPS 2019]

Reviewer 1



Overall this is a really interesting idea incorporating concrete visual concepts and more abstract metaconcepts in a joint space and using the learning of one to guide the other. There are some issues below, mostly details about training implementation, that could clear up my questions. 1. Why not use pretrained word embeddings for the GRU model? The issue here is that the object proposal generator was trained on ImageNet, meaning it almost definitely had access to visual information about the held out concepts in Ctest. The GRU baseline, even signficantly less training data, outperforms for instance-of. Would the same be true of synonym metaconcept if it had been trained on data like Common Crawl or Wikipedia? The authors should either pretrain the vision model with access only to the training concepts or train word embeddings with more data for a more fair comparison. 2. In Section 4.3.2, how do the authors know instance-of metaconcept is the reason the model outperforms the others on CLEVR? Also, all the models essentially have the same performance on GQA. 3. In Table 4, high performance on synonym makes sense because the concepts are visually similar and visual features are computed from a model pretrained on ImageNet. For the instance-of metaconcept, which likely relies more on linguistic information than synonym, the GRU with word embeddings trained on a small amount of data outperforms the pretrained semantic parser. There’s a huge performance gap between the metaconcepts, the largest being for this model, that should be explored more in-depth. This also ties into the comment above. 4. Did authors consider using existing zero-shot compositional visual reasoning datasets (e.g. [1]) instead of manually generating questions and answers? Clarity: - Well-written paper with clear, easy-to-follow sections. A small change on Fig 3 could be to add more space between the lines coming from the meta-verify parse. - Was the GRU portion of GRU-CNN also pretrained only on the questions? - Number of metaconcepts used isn’t clear until the final section before the conclusion (except brief mention in Table 1). Perhaps add an additional line in dataset section. - How was the semantic parser trained? If it wasn’t on just training questions, in the same way the GRU was trained, then this isn’t a fair comparison with the GRU and GRU-CNN baselines. - Table 4 only includes results for GQA but also mentions results for CLEVR in the text. Notes: -line 110: “In Figure 2, We illustrate..” -> “In Figure 2, we illustrate” -line 119: “we first runs..” -> “we first run” -line 181: “we assume the access to” -> “we assume access to: -line 247: “it fails the outperform” -> “it fails to outperform” -line 260- “to indicate the models” -> “to indicate to the models” References: [1] C-VQA: A Compositional Split of the Visual Question Answering (VQA) v1.0 Dataset. Agrawl et.al. 2017.

Reviewer 2



Strengths: - The proposed model is relevant and close to how most human learning is done ie. through hierarchy of concepts and often inference is done through interaction of those concepts. - The proposed model learns the visual concepts and the meta concepts which specifically aid in zero shot learning and generalizing to noisy, unseen and even biased dataset conditions.

Reviewer 3



originality: it is difficult to evaluate the originality because there is no discussion of relevant work on metaconcepts. Perhaps none exists (I'm not familiar with the area), and if so the authors should be more clear about that. quality: I find the discussion of the results in Table 3 puzzling -- it looks to me like the metaconcept "instance of" is (barely) helping in the synthetic dataset only, and is not providing any benefit in the real dataset. Moreover, these accuracy are awfully close to 50% -- is this not chance accuracy? It would be helpful to provide significance testing for the differences. clarity: Generally, the paper is well written, though it does assume a bit of the reader in terms of background knowledge. Also, L216 mentions that the proposed method will be compared to BERT but I cannot find this comparison. significance: Because only two metaconcepts are tested, and only one of them actually somewhat benefits concept learning in a real dataset, it is difficult to say how significant this method is. *** POST-REBUTTAL *** I had 2 major concerns about this work. The first was that the considered metaconcepts were quite contrived, and it was not easy to see what other metaconcepts can be incorporated. In their rebuttal, the authors include results with a new metaconcept "hypernym", but I fail to see how this is different from their original metaconcept of "is instance of". The second major concern was that the experimental results did not seem thorough -- I can't find any results reported over multiple random seeds, despite the authors claiming that they have provided clear error bars and standard deviations in the reproducibility checklist. This concern also holds for the new results reported in Tables A,B, and C. Specifically the results in Table C would have been more convincing if they had been computed in a bootstrapped fashion (if they had actually been done this way, the authors should have provided a standard deviation). These concerns still stand after the rebuttal. Additionally, a more minor concern is that I feel this work is not a good fit for the Neuroscience and Cognitive Science track that it has been submitted to. I find it very loosely related to human psychology, and difficult to see how a researcher in this area (like myself) would be interested in this work. For these reasons, I maintain my review score.

[Author Response · NeurIPS 2019]

We thank all reviewers for their insightful and constructive comments. We'll release all code and data.

**(R1) GRU with pre-trained embeddings and BERT.** We supplement extra experiments using GRU with pre-trained
GloVe embeddings and BERT in Table A, as an extension to the Table 4 in the paper. Pre-trained embeddings from large
language corpus indeed help baselines better predict the *synonym* relation between unseen concepts, but our VCML
still outperforms them. We also observed that pretrained text embeddings do not improve the GRU baseline on the
*instance-of* tests, whose weights are pretrained by language modeling on training questions.

Table A: Visual grounding helps predict metaconcepts between unseen concept pairs (evaluated in metaconcept QA accuracy).

| CLEVR | Q. Type | GRU | GRU (GloVe) | BERT | VCML | GQA | Q. Type | GRU | GRU (GloVe) | BERT | VCML |
|---|---|---|---|---|---|---|---|---|---|---|---|
| Synonym | 50.0 | 55.1 | 56.6 | 80.7 | **86.3** | Synonym | 50.0 | 53.4 | 71.5 | 76.0 | **94.5** |
| InstanceOf | 25.0 | 58.9 | 42.5 | 44.6 | **72.2** | InstanceOf | 12.5 | **26.3** | 14.6 | 14.9 | 19.9 |

**(R1, R3) The 'instance-of' metaconcept in de-biasing (Sec. 4.3.2, Table 3).** To verify that the *instance-of* meta-
concept helps de-biasing, we perform an ablation study on VCML with and without *instance-of*. On CLEVR, VCML
performs better with the instance-of metaconcept (55.6% *vs.* 43.3%). Meanwhile, we speculate that models on GQA
de-biasing perform similarly since GQA concept categories are not well reflected in vision, especially for categories
defined by functionality instead of appearance, such as vehicles. To evaluate VCML on interpreting metaconcepts better
associated with visual appearance in natural images, we have supplemented results on CUB[*] (see below L26–L31).

**(R1) Instance-of metaconcept generalization in Table 4.** We agree with the review on that linguistic information
helps more on the instance-of metaconcept in GQA, and the supplementary results in Table A also support this. Since
VCML learns concept embeddings completely from visual data, it performs worse than the linguistic baselines.

**(R1) Zero-shot compositional visual reasoning.** Thanks for the suggestion. In this work we use manually generated
datasets for two reasons. First, they enable controlled and diverse experiments such as de-biasing. Second, the extra
metaconcept questions are essential: the de-biasing generalization will be otherwise ill-formed.

**(R1) Technical details.** The word embeddings in GRU-CNN is pretrained on the question set, same as in GRU. The
semantic parsers used by NS-CL and our VCML are identical, both trained on question-program pairs. The fact that
VCML outperforms the metaconcept-agnostic NS-CL suggests the importance of metaconcept learning.

**(R2) Unsupervised discovery of metaconcepts.** We agree that the unsupervised discovery of metaconcepts is a
promising direction[†]. The main contribution of this paper is to incorporate metaconcepts into visual concept learning,
in the form of supplementary question-answer pairs. The extra information enables learning from less and even biased
data, which is ill-formed if no extra supervision (e.g., human-designed metaconcept-related questions) is present.

**(R2, R3) Generalizing to new concepts and metaconcepts.** We also apply VCML on the
CUB dataset[*] to learn the *hypernym* metaconcept from visual data. Data are generated from
the biological taxonomy of birds. We train different models on a partial set of the taxonomy,
by providing the hypernym relationship of ∼74K pairs between 273 concepts, and evaluate
them on ∼9K pairs between 93 concepts in the held-out set. Shown in Table B, our model
outperforms both visual and linguistic baselines, which supports the generality of our VCML.

Table B: Metaconcept generalization evaluation of *hypernym* on CUB.

| Model | Acc. (%) |
|---|---|
| Q. Type | 50.0 |
| GRU (Lang.) | 74.3 |
| BERT | 73.1 |
| GRU-CNN | 76.7 |
| NS-CL | 54.3 |
| VCML | **85.5** |

**(R3) Originality of metaconcept learning.** This paper introduces a new approach to si-
multaneous learning of visual concepts and metaconcepts. Moreover, its applications such
as de-biasing with metaconcepts have never been addressed before. Existing research on
metaconcepts have been mostly restricted to linguistic domains. Two related topics are visual
compositional learning and knowledge graph completion, both discussed in Section 2.

**(R3) Generality of metaconcept operators.** Our design of metaconcept operators is inspired by TransE[‡], a frame-
work for linguistic knowledge graph embeddings. It is a general operator for metaconcepts/relations between concepts.

**(R3) Connection to Platanios et al.** Thanks for suggesting the related work, which we will cite and discuss. In
VCML, the projection embedding transforms the object embedding into a subspace, in which a cosine similarity is
then computed to classify the object. This differs from the projection network of Platanios et al. which transforms a
language embedding into a neural network parameter for encoding input sentences.

**(R3) Application of concept embeddings to downstream tasks.** We supple-
ment extra results on the CLEVR visual reasoning challenge (Visual QA) and a
referential expression task. The task of referential expression is to select out a
specific object from a scene given a description (e.g., the red cube). We compare
VCML with and without metaconcept information using the QA accuracy for visual
reasoning and Recall@1 for referential expressions. Table C suggests that the meta-
concept information significantly improves visual concept learning in low resource
settings, using only 10K or even 1K visually grounded questions.

Table C: Evaluation of the learned concept embeddings on visual reasoning (in QA accuracy) and referential expression interpretation (in Recall@1).

| | #Train | w/. | w/o. |
|---|---|---|---|
| Visual QA | 10K | 74.8 | 73.6 |
| | 1K | 65.7 | 61.0 |
| Ref. Expr. | 10K | 71.2 | 70.2 |
| | 1K | 55.0 | 51.2 |

[*]Wah et al. The Caltech-UCSD Birds-200-2011 Dataset. TechReport, Caltech, 2011.
[†]Kemp and Tenenbaum. The Discovery of Structural Form. PNAS 2008
[‡]Bordes et al. Translating Embeddings for Modeling Multi-Relational Data. NeurIPS 2013

[Meta-Review · NeurIPS 2019]

The authors present a joint framework for acquiring both visual concepts of objects and linguistic metaconcepts describing relationships between the visual concepts in visual reasoning tasks (images paired with question-answer pairs), and demonstrate that this works on synthetic and real-world image datasets. Part of the novelty of this work is in incorporating the metaconcepts into visual concept learning, and the proposed model somewhat mirrors how human learning is done. The approach to concept learning aids in zero-shot learning. Reviewers would like to see more careful and thorough experimental validation, and are concerned that the metaconcepts are not realistic.